# Effect of Morphologically Controlled Hematite Nanoparticles on the Properties of Fly Ash Blended Cement

**DOI:** 10.3390/nano11041003

**Published:** 2021-04-14

**Authors:** Pantharee Kongsat, Sakprayut Sinthupinyo, Edgar A. O’Rear, Thirawudh Pongprayoon

**Affiliations:** 1Department of Chemical Engineering, Faculty of Engineering, King Mongkut’s University of Technology North Bangkok, Bangkok 10800, Thailand; pantharee99@gmail.com; 2Center of Eco-Materials and Cleaner Technology, King Mongkut’s University of Technology North Bangkok, Bangkok 10800, Thailand; 3Siam Research and Innovation Co., Ltd., Saraburi 18260, Thailand; sakprays@scg.com; 4School of Chemical, Biological and Materials Engineering and Institute for Applied Surfactant Research, University of Oklahoma, Norman, OK 73019, USA; eorear@ou.edu

**Keywords:** hematite nanoparticles, fly ash blended cement, cement hydration, compressive strength, workability

## Abstract

Several types of hematite nanoparticles (α-Fe_2_O_3_) have been investigated for their effects on the structure and properties of fly ash (FA) blended cement. All synthesized nanoparticles were found to be of spherical shape, but of different particle sizes ranging from 10 to 195 nm depending on the surfactant used in their preparation. The cement hydration with time showed 1.0% α-Fe_2_O_3_ nanoparticles are effective accelerators for FA blended cement. Moreover, adding α-Fe_2_O_3_ nanoparticles in FA blended cement enhanced the compressive strength and workability of cement. Nanoparticle size and size distribution were important for optimal filling of various size of pores within the cement structure.

## 1. Introduction

Nanomaterials have been widely studied for their beneficial effects on the properties of cement and concrete. Nanoparticles of silica, titanium dioxide, hematite, alumina, clay and other substances increase the mechanical strength and durability of concrete [1,2]. The mechanisms underlying these improvements rely on structural characteristics of the nano-modifier. Increased surface area accelerates the pozzolonic reactions while the smaller size facilitates densification on filling voids. This means the method of synthesis and control of structure of the nanoparticles is important.

Cement consists mainly of Ordinary Portland Cement (OPC) as calcium oxide (CaO) [3,4]. Partial replacement of OPC with fly ash (FA) can reduce CO_2_ emissions and help address an environmental waste problem of the FA byproduct from coal combustion in electrical power production [5,6]. Moreover, FA blended cement has been found in actual use to improve workability, strength, durability and hardened cement composites [7,8,9].

The hardening of cement materials is the result of hydration reactions of OPC with water. Basically, hydration can be separated into five stages (Appendix A) [4,10,11,12,13]. The first stage, very soon after mixing, involves C_3_A reacting with water to form an aluminate-rich gel. A few minutes later, one observes low heat evolution in the induction period, when C_3_S and C_2_S in the cement start to react and form a C-S-H gel. For several hours, early formed ettringite converts to CAS^−^, allowing continuation of the C_3_A hydration process. As C_3_A is hydrating within the cement paste, reaction species create a shoulder in the decelerating rate region of the peak. Lastly, the C_4_AF reacts in a similar manner as C_3_A but more slowly [13,14,15,16].

Many researchers have demonstrated the enhanced properties of OPC with added iron oxides. Hematite (α-Fe_2_O_3_) enhances cement hydration, increases compressive strength and lowers drying shrinkage [17,18]. The added hematite nanoparticles act in part by filling pores of the cement mortar [19,20]. The α-Fe_2_O_3_ also acts as a foreign nucleation site that accelerates the formation of C-S-H gel as the hydrated product of crystalline Ca(OH)_2_, especially at stage 3 hydration, leading to greater cement strength [13,15]. As noted above, the size of a nucleation particle can change the microstructure of hydrated cement and much improve its mechanical strength. This work is aimed at understanding how properties of nano-hematite affect the hydration, workability and strength of FA blended cement.

Researchers have described several approaches to the synthesis of α-Fe_2_O_3_ nanoparticles including coprecipitation [21], hydrothermal [22] and surfactant-assisted hydrothermal methods [23]. Of these techniques, synthesis of nanoparticles using a surfactant template offers some advantages. Especially, the property of a surfactant to form micellar aggregates in solution at its critical micelle concentration (CMC) provides a means to control size and structure of the nanoparticles [23,24]. The structure, size and other properties of α-Fe_2_O_3_ nanoparticles are affected by the shape and size of the micelles, which are determined by the chemical structure of the surfactant molecule.

The function and mechanisms of α-Fe_2_O_3_ on FA blended cement have yet to be established in the literature. Research is needed to develop the fundamental mechanisms relating hydration and porosity to macroscopic observations of mechanical strength of the composite upon hematite nanoparticle addition. This study focused on the effects of α-Fe_2_O_3_ of various sizes on the hydration reaction and workability of FA blended cement. Three structure-controlled hematite nanoparticles have been synthesized by the surfactant-assisted method using anionic, cationic and nonionic surfactants and each characterized by X-ray diffraction (XRD), Transmission electron microscope (TEM) and Field-emission-scanning-electron-microscopy (FE-SEM). FA blended cements were prepared with each of the three synthesized α-Fe_2_O_3_ nanoparticles as well as with a commercial α-Fe_2_O_3_ nanoparticle. Heat flow calorimetry was used to follow cement hydration, while SEM was used for microstructural analysis with the aim of understanding morphological structure for optimal performance. The workability during hydration, an important behavior for utilization of cement composites, was also evaluated. Strength and workability were determined following standard tests for fresh cement.

## 2. Materials and Methods

### 2.1. Materials

Ordinary Portland Cement (OPC) type 1 and fly ash were provided by cement plant (SCG Cement Co., Ltd., Saraburi, Thailand). Commercial α-Fe_2_O_3_ came from a vendor for cement factory and its properties were analyzed by XRD and SEM. The synthesized α-Fe_2_O_3_ nanoparticles were prepared by surfactant-assisted method as described previously [23]. Iron (II) sulfate heptahydrate (FeSO_4_·7H_2_O, 99%) from Quality Reagent Chemical (QREC, Auckland, New Zealand) served as the precursor for the nanoparticle synthesis. Ammonium hydroxide solution (NH_4_OH, 30 wt%) in water was purchased from Sigma–Aldrich (Sigma-Aldrich In., Saint Louis, MO, USA). Sodium dodecyl sulfate (SDS) was obtained from Ajax Finechem (Ajex finechem Inc., Taren Point, Australia). Cetyltrimethylammonium bromide (CTAB) provided from Amresco (Amresco Inc., Solon, OH, USA). Polyoxyethlyene tert-octylphenyl ether (TX100) was supplied from Applichem Panreac (Applichem Gmbh Inc., Darmstadt, Germany). All chemicals for α-Fe_2_O_3_ synthesis were used as-received.

### 2.2. Synthesis and Characterization of α-Fe_2_O_3_

In this case, α-Fe_2_O_3_ nanoparticles were synthesized by the surfactant-assisted method to provide morphological control. Three surfactants (SDS, CTAB and TX100 as anionic, cationic and nonionic surfactants, respectively) were used in this study. For each surfactant, an aqueous solution (200 mL distilled, deionized water) at twice the CMC (8.40 mM for SDS, 0.98 mM for CTAB and 0.24 mM for TX100) was prepared. The surfactant solution was stirred at 60 °C for 1 h until it was clear. Stirring continued for 2 h after addition of precursor iron (II) sulfate heptahydrate (20 g). After that 30 wt% ammonia solution (5 mL) was carefully added into the solution with continuous stirring for another 2 h. The mixture color changed from light yellow to dark blue as the hematite nanoparticles were formed in the solution. Lastly the nanoparticles were filtered and washed with distilled water for removing the excess surfactants and other impurities. The filtrate was dried at 80 °C in an oven for 24 h. Dried nanoparticles were further calcined at 600 °C for 4 h under air for crystallization.

X-ray diffraction (X-Ray Diffractometer; Bruker AXS Model D8 Discover) was used for crystal structure and phase analysis by CuK_α_ radiation (λ = 1.5406 Angstrom) with a scanning speed of 0.02 step/s in 2θ ranging from 10 to 70 degree. A transmission electron microscope (TEM), Philips-TECNAI 20, was used to observe the nanoparticle morphology as 2D. Field-emission-scanning-electron-microscopy (FE-SEM), JSM-6480LV, was also used for morphological observation of the synthesized nanoparticles and FE-SEM micrographs were used to observe the nanoparticle morphology as 3D and evaluate the particle size by ImageJ software.

### 2.3. FA Blended Cement Preparation

Seven samples were prepared by varying the compositions of binders (Table 1). The basic binders of this study included OPC and FA. In this case, α-Fe_2_O_3_ were varied in both quantity and quality. The percentage of FA was fixed at 40% by weight due to its high performance in cement mixtures designed for both high strength and high durability [5,9,25,26]. An increasing mass percentage of α-Fe_2_O_3_ nanoparticles was replaced with a decreasing percentage OPC.

For hydration and setting time investigation, samples were prepared as cement pastes with the same compositions shown in Table 1. For slump test, flow table and compressive strength measurements, the samples were constituted as the mortar by adding fine aggregate sand in the size range of 2.0–2.8 mm, with the mass ratio specified for the slump test experiment. Sample codes are indicated in Table 1.

### 2.4. Hydration Analysis

The heat evolution emitted from cement hydration was observed by isothermal calorimetry (TA instrument, TAM Air 8 channel). The calorimeter gauges heat flow associated with physical processes and chemical reactions for observation of the hydration stages and heat reaction rate, respectively. In this work, a twin-type calorimeter was used with separate sample and reference chambers.

The behavior of cement hydration can generally be divided into five stages (Appendix A). In line with previous work from Yuenyongsuwan et al. [12] and Kim et al. [27], this study concentrated on the acceleration period (stage 3) and the deceleration period (stage 4) of the hydration behavior curve. The effect of the nanofiller on hydration can be observed at the maximum heat flow values which occur in these stages [14,27]. Binders tested included mixtures of OPC, fly ash and α-Fe_2_O_3_ nanoparticles (Table 1). The total binder (40.0 g) was mixed and stirred in the cup for 40 s, then 5.0 g of binder was weighed and carefully filled into a testing glass bottle for the calorimetry test. Water (3.8 g) in the syringe that was placed in the calorimeter was slowly combined with the binder for hydration observation (Appendix A). All measurements were conducted over 40 h to cover the hydration period.

### 2.5. Workability and Compressive Strength

#### 2.5.1. Setting Time

Setting time, the required time for stiffening of cement paste, was investigated by the Vicat needle method, ASTM C 191 standard. The cement pastes for all samples of this test were prepared to a total mass of 650 g. Water and binder (w/b) at a mass ratio of 0.3:1 were put in the mix pot while continuously stirring at low speed (140 ± 5 rpm/min) for 30 s and followed by continuous stirring at medium speed (285 ± 10 rpm/min) for 60 s. A mold (an internal diameter of 70 mm at the top and 80 mm at the bottom and a height of 40 mm) was filled with the cement paste within 5 min after mixing. The bearing surface of the needle was brought into contact with the cement paste, the scale zeroed, and the plunger immediately released with the needle allowed to settle for 30 s when depth of penetration was recorded. The measurement was repeated every 15 min until a penetration of 25 mm was reached as indicated by the scribe mark. The time at which cement starts to harden and completely loses its plasticity is called initial setting time, while the time of the change from the plastic state to solid state is called the final setting time.

#### 2.5.2. Mini-Slump Test and Flow Table Test

The mini-slump test, according to ASTM C-143 (ASTM 2004), was used to measure the mortar behavior under the action of gravity in a compacted inverted cone or mold for testing. The mini-slump cone has top and bottom diameters of 60 mm and 70 mm, respectively, with a cone height of 100 mm. The mortar for each sample (Table 1) was prepared to a total mass of 500 g. Water–binder (w/b) and binder-sand mass ratios were fixed at 0.45:1 and 1:3, respectively. Firstly, binder, sand and water were put into the mix pot and then continuously mixed at 140 rpm/min for 3 min. Next, the mixture was filled into the mini slump cone that had been placed on a table. The inverted cone mold was slowly lifted up and the height of the mortar placed on the table was measured and recorded (Appendix A).

The flow table (according to ASTM C1437) was also used to evaluate workability. The mortar sample and cone mold were similar to that in the slump test. After the cone mold was raised, the mortar sample was dropped 25 times from the height of the cone mold, approximately 100 mm, within 15 s. Then the diameter of the spread mortar on the table was measured and recorded.

#### 2.5.3. Compressive Strength

The compressive strength of mortar samples was tested according to the standard ASTM C109/C109M. The mortar samples were prepared by mixing binder and fine aggregate in a 1:3 mass ratio. The w/b ratio of mass was fixed at 0.45:1. A total 200 g binder and 600 g fine aggregate were mixed together under dry condition for 1 min and then water was added. The mortar was cast in a mold with the dimension of 50 × 50 × 50 mm. When casting was completed, the mortar was left in the mold for 24 h while curing at 25 °C. After that, the mortar specimen was removed from the mold and placed in water at 25 °C. The specimen was used to test the compressive strength at 7-, 28- and 90-days curing. The test at 90 days curing was of interest for further application in self-compacting concrete.

### 2.6. Morphology of Cement Paste

A scanning-electron-microscope (SEM), Quanta-450, was used for morphological observation of the FA blended cement paste specimens at 7-, 28- and 90-days curing. The specimens were prepared by mixing water and binder as shown in Table 1 with w/b ratio at 0.3:1. After mixing, the cement pastes were put into plastic tubes. After 24 h, the plastic tubes were peeled off and the cement specimens were soaked in water for curing for 7, 28 and 90 days at 25 °C.

## 3. Results and Discussion

The purpose of this study was to evaluate the performance of fly ash cement modified with a series of synthetic α-Fe_2_O_3_ with particle sizes on the order of 10–195 nm. Knowledge gained with different α-Fe_2_O_3_ nanoparticles can be used to improve fly ash blended concrete.

### 3.1. Synthesized α-Fe_2_O_3_ Characterization

In this case, α-Fe_2_O_3_ nanoparticles were characterized by XRD, TEM and FE-SEM to determine phase, size and shape of the synthesized nanoparticles.

XRD patterns of α-Fe_2_O_3_ are depicted in Figure 1. The recorded and indexed diffraction patterns of the sharp peaks are as expected for highly crystalline samples. The results show peak positions of 2θ values at 23.88°, 33.47°, 36.25°, 41.54°, 49.58°, 54.82°, 57.55°, 63.44° and 64.67°, indexed as (012), (104), (110), (113), (024), (116), (018), (214) and (300) planes that fit the standard pattern of α-Fe_2_O_3_ [28]. No diffraction peaks corresponding to other phases were present, indicating a high purity of α-Fe_2_O_3_ for all synthesized samples. These clearly indicate the formation of fully crystalline iron oxide of α-Fe_2_O_3_ structure without other iron oxide phases.

The crystallite size was calculated from XRD patterns using Scherrer equation and applying full-width half-maximum (FWHM) of characteristic peak (at 104 or 2θ = 33.3°) of α-Fe_2_O_3_ with the following equation:(1)$$Crystallite\;size=\frac{0.9\lambda }{FWHMcos\theta }$$
where *λ* is the X-ray wavelength (1.5406 Å in this study) and *θ* is the diffraction angle for the (104) plane. The crystallite sizes of the synthesized α-Fe_2_O_3_ using SDS, CTAB and TX100 at 2 CMC were approximately 20.8, 35.5 and 26.3 nm diameters, respectively, whereas the crystallite size of commercial α-Fe_2_O_3_ was approximately 29.8 nm. Basically, crystallization occurs in two major steps, the first step is nucleation from the reaction and then crystal growth to increase the size of particles, leading to a stable crystalline state. For the second step, an important feature may occur due to the surfactant micelle with crystal defects themselves appearing as open inconsistencies such as pores and cracks [29]. These results for the crystallite size show some correspondence to the micelle size of SDS (3.5–4.0 nm) [30], CTAB (118.0–192.0 nm) [31] and TX100 (10.2 nm) [32].

Both TEM and FE-SEM were used to observe the agglomerate particles in this study. Grain size was measured by TEM and the particle morphology was determined by FE-SEM.

The 2D images of TEM in Figure 2 clearly show that the individual α-Fe_2_O_3_ were rounded as the sphere-like shape with various curvatures in all samples. The grain size of 2D images in Figure 2 were analyzed by ImageJ analysis software. A single grain within a nanoparticle corresponds to a crystallite, while multiple aggregated grains exist within most nanoparticles. 3D images of FE-SEM micrographs and ImageJ analysis software were used for morphological observation and size estimation of all samples (Figure 3). Results confirmed the spheroidal shape of all samples and relative sizes depending on surfactant used. The smallest ones occurred when using SDS for the synthesis, whereas CTAB yielded the largest nanoparticles. The average grain size and average particle size of the α-Fe_2_O_3_ synthesized via SDS were 54.1 nm and 65.2 nm. Grain sizes ranged between 10 and 110 nm while particle sizes varied from 10 to 130 nm. The grain size is smaller than particles size because one particle can have several grains. The average grain sizes of synthesized α-Fe_2_O_3_ using CTAB and TX100 were different at 100.6 nm and 79.8 nm, even though particle sizes were similar. Particle sizes were 131.0 nm and 122.0 nm average diameters, respectively, in a similar particle size range of 35 to 195 nm. These compared to an average particle size for the commercial α-Fe_2_O_3_ nanoparticles of approximately 174.1 nm with particle sizes ranging from 100 to 240 nm.

The relative dimensions of the surfactant-synthesized particles from TEM and SEM confirmed the XRD crystal size estimation. Others have found that the size of nanoparticles could be controlled by micelles in water, often with increasing particle size for larger micelles [29,30,31]. Summarized results of crystal size and particle size are shown for comparison in Figure 4. Results confirmed that surfactants significantly influence the shape and size of the nanoparticles, in accord with the findings of Jing et al. [33] and Colombo et al. [34].

### 3.2. Hydration of FA Blended Cement: Effect of Amount and Types of α-Fe_2_O_3_ Addition

The hydration mechanisms of FA blended cement with added α-Fe_2_O_3_ nanoparticles were studied by calorimetry, commonly used to assess acceleration (stage 3) of the hydration reactions [16]. Normally, FA acts to reduce the heat of hydration as indicated by a lowering of the highest peak for OPC or pure cement. In contrast, it has been reported that α-Fe_2_O_3_ improves the rate of heat generation depending on both the particle size distribution and quantity of α-Fe_2_O_3_ [13,16]. Thus, the effect of added hematite to an FA blend is of interest.

Results for the FA blended cements without and with α-Fe_2_O_3_ up to an age of 40 h are shown in Figure 5 and Figure 6. The general shape of the calorimetry curves is typical for cement hydration. A quick heat flow release in stage 1 (0 to 1 h) occurs due to the neutralization of electrostatic charge on the particle surfaces, dissolution of calcium sulfate and alkali sulfates. Stage 2 (1 to 3 h) starts with the onset C-S-H crystallization, precipitating a layer around the cement particles, which slows down the rate of hydration reactions with formation of a barrier around the particles. Stage 3 (3 to 10 h) proceeds with the main C-S-H gel creating an outer shell around the particles with high heat flow release. This is the acceleration period of the reaction. After peak heat flow release, the reaction rate of C-S-H gel formation decreases during the deceleration period from 10 to 40 h (stage 4).

With added commercial α-Fe_2_O_3_, the maximum rate of heat release shifts to earlier times while the maximum value decreases as the level of nanoparticles goes from 1.0 to 5.0% (Figure 5). At 1.0% α-Fe_2_O_3_ nanoparticles (C1), the time period of C_3_S production from hydration was altered with the peak maximum appearing at 9.3 h, faster than without added α-Fe_2_O_3_ by approximately 0.5 h. This showed that the addition of α-Fe_2_O_3_ nanoparticles advanced the initiation and end of the acceleration period of cement hydration, resulting in the increase of the C–S–H crystal growth as reported by Kiamahalleh et al. [35]. After this enhanced reactivity, cement hydration slowed down in the deceleration period of stage 4 [14,36]. Maximum heat release of the α-Fe_2_O_3_ added FA cement decreased with the amount of the nanoparticles, becoming lower than that of FA cement alone. These findings imply that α-Fe_2_O_3_ nanoparticles acted as an accelerator to form C-S-H gel [37,38] with less exothermic heat overall being released as a result of reduced formation of C-S-H gel and ettringite [11,18]. Kishar et al. [39] similarly found that 1.0% α-Fe_2_O_3_ nanoparticles added in modified cementitious materials enhanced the cement hydration rate.

How the various surfactant-synthesized α-Fe_2_O_3_ nanoparticles affected the hydration of FA blended cement is shown in Figure 6. Peak heat flow generation observed for the 1.0% α-Fe_2_O_3_ added FA cement samples was approximately 0.0026 W/g at 9.3 h for the SDS synthesized nanoparticles, 0.0027 W/g at 9.1 h for the CTAB synthesized nanoparticles, and 0.0027 W/g at 9.2 h for TX100 synthesized nanoparticles. These results reflect the effect of size and size distribution since the type of surfactant influenced these characteristics of the α-Fe_2_O_3_ nanoparticles. For comparison, peak heat flow generation for the FA control sample was 0.0025 W/g at 9.8 h while the 1.0% commercial α-Fe_2_O_3_ value (C1) of 0.0026 W/g occurred at 9.3 h. A relationship can be seen between the heat of hydration and particle size for the FA blended cement with added hematite. The lowest heat of hydration occurred with the smallest nanoparticles synthesized (SDS, 10–130 nm) while the larger particles and broader size distribution of the nanoparticles synthesized with CTAB and TX100 (35–195 nm) had higher maximum heat flow values.

When α-Fe_2_O_3_ was added to FA blended cement, the maxima in the acceleration period occurred sooner at 9.1–9.4 h compared to 9.8 h in the absence of added α-Fe_2_O_3_. This indicated that the α-Fe_2_O_3_ nanoparticles function as hydration catalyst or accelerator, depending on the percentage added and particle diameter. The FA blended cement with added α-Fe_2_O_3_ nanoparticles, synthesized using CTAB and TX100, had the earliest maxima heat flow values along with their bigger particles and broader particle size distribution (35–195 nm), whereas the FA blended cement with added α-Fe_2_O_3_ nanoparticles, synthesized using SDS, had the latest maximum according to the smallest particles (10–130 nm). Kayali et al. [13,14] and Kocaba et al. [15] reported similar observations.

How the various sizes of the α-Fe_2_O_3_ nanoparticles synthesized using CTAB and TX100 advanced the acceleration period of cement hydration (higher heat of hydration) can be explained by greater nucleation to yield more C-S-H gel [40]. Several nucleation sites of C-S-H gel can produce a branch-like nanostructure, interconnected via electrostatic and van der Waals forces. This structural aspect of the C-S-H gel represents a scaffolding component enhancing mechanical properties of hardened cement. The CTAB and TX100 synthesized hematite nanoparticles had a broad range of sizes causing a different morphology of C-S-H and granular agglomeration of C-S-H gel as suggested by the slightly higher peak of the heat evolution (0.0027 W/g) compared to that of the commercial hematite-added FA blended cement (0.0026 W/g). The comparison of the highest peak hydration in the relation of time and heat flow of all studied samples (Figure 5 and Figure 6) was concluded in Appendix A.

Particles the size of α-Fe_2_O_3_ nanoparticles can also affect packing density of cementitious materials [41,42]. At the nanoscale (on the order of 10–100 nm), α-Fe_2_O_3_ nanoparticles can contribute to higher packing density and lower water content in the scaffolding component so the hydration reaction proceeds more slowly [41,42,43]. At the mesoscale with bigger particles and a broader particle size distribution (on the order of 100 nm), high water content in the scaffolding component leads to an increased hydration reaction [41,42,43]. This knowledge implies that cement-based materials could be further upgraded by fine-tuning the nano and microstructure [41].

### 3.3. Workability and Compressive Strength of FA Blended Cement: Effect of α-Fe_2_O_3_ Addition

Workability, a property of freshly mixed FA blended cement directly tied to strength and appearance, is typically gauged by setting time, mini-slump test and flow table test, though attempts have been made to relate workability directly to the hydration reaction. Increased hydration products or C-S-H gel correspond to decreased excess water and improved workability. In practice, it is necessary to place and consolidate the cement product before initial setting starts and then not to disturb the sample until the final setting of mortar or concrete. If delayed, the cement will lose strength.

#### 3.3.1. Setting Time

Setting times of cement paste, related to workability, are defined as of the moment cement changes from liquid state to plastic state and then to solid state. Two important properties, initial and final setting times, indicated the stiffness of the cementitious matrix and the rate of solidification of mortar or concrete with added nanoparticles.

Setting times were observed for all samples as shown in Figure 7. The initial and final setting times of FA blended cement control were 4.5 and 6.0 h, respectively. Normally, the initial and final setting times of cement are specified as a requirement of related standards (EN-197, 2011; ASTM C150, 2015). For the initial setting time, the value should not be less than 0.5 h while the final setting time should not be greater than 10.0 h. This means that the FA blended cement must be in place within 4.5 h after mixing with water before it will consolidate and then must be left undisturbed until 6.0 h.

When commercial α-Fe_2_O_3_ nanoparticles were added at 1.0% in the FA blended cement, the initial setting time and the final setting time decreased to 3.4 h and 5.6 h. FA blended cement with the hematite nanoparticles synthesized using CTAB gave the lowest initial setting time at 3.1 h in agreement with its highest rate of hydration reaction (shown in Figure 5 and Figure 6). The slightly shorter initial setting time of the FA blended cement with the added hematite nanoparticles led to higher strength gain from the effect of nucleation on high degree of hydration, low porosity and permeability [42]. For the final setting times of α-Fe_2_O_3_ modified samples, the FA blended cement with CTAB nanoparticles gave the highest final setting time at 5.9 h, while the FA blended cement with the SDS nanoparticles exhibited the lowest final setting time at 5.5 h. Final setting times for all samples met the applicable standards of 10.0 h or less.

#### 3.3.2. Mini-Slump Test and Flow Table

The mini-slump test measures the workability of fresh mortar from its behavior under the action of gravity while the flow table test reflects the consistency or wetness of OPC, giving another measure of the workability. Results for these tests are presented in Figure 8. The mini-slump test of control sample FA blended cement exhibited the greatest height at 6.7 cm, while the FA blended cement containing α-Fe_2_O_3_ synthesized with CTAB showed the least height at 5.1 cm. The trend of the data from the flow table test was the same as that for the mini-slump test results. The flow table test showed that the FA blended cement with hematite nanoparticles synthesized via CTAB had the lowest spread at 22.0 cm, while the FA blended cement was the highest at 34.5 cm.

Results of the mini-slump test and flow table of the FA blended cement control were higher than those of all α-Fe_2_O_3_ added FA blended cements. As the amount of α-Fe_2_O_3_ was increased, values obtained for flow test also increased. These findings with the heat flow results show workability depends upon the water inside the cement during hydration. Water exists in four forms including capillary water, adsorbed water, interlayer water and chemically combined water. In general, capillary water, or generally called “free water”, affects the flow of cement with the attractive forces exerted by the solid surface [43,44]. When the nanoparticles of α-Fe_2_O_3_ were added, values of the mini-slump test and flow table for the cement composites slightly decreased due to the displacement of free water caused by the reduction of interstitial volume inside the cement during reaction [45]. Currently the α-Fe_2_O_3_ nanoparticles are used in concrete work to reduce the amount of free water by accelerating the hydration reaction [46].

#### 3.3.3. Compressive Strength

The compressive strength of a cured cement composite is commonly measured at 7, 28 and 90 days. As expected, the compressive strength increased with time (Figure 9). Compressive strength of the FA blended mortar control rose with the value at 7 days being 62% of that at 90 days. This can be compared to the work of Abd elaty et al., who found that a cement composite reaches as much as 60% of the equilibrium value for all mechanical properties at 7 days [47]. At 7 days, the compressive strength of FA blended mortar with commercial hematite 3.0% was the lowest at 24.0 MPa, while FA blended mortar with CTAB-synthesized α-Fe_2_O_3_ or TX100-synthesized α-Fe_2_O_3_ of various sizes was the highest at 26.0 MPa. At 90 days, the same CTAB mortar still had the highest compressive strength, now at 47.1 MPa, with a similar value for the TX100 synthesized nanoparticle modified mortar.

Higher compressive strength among the α-Fe_2_O_3_ modified samples corresponded to faster rates of the hydration reactions. The highest compressive strength occurred with the variable nanosize of the CTAB-assisted synthesized α-Fe_2_O_3_ in line with the hydration behavior within the various voids in the mortar. The results imply that the α-Fe_2_O_3_ nanoparticles function as an accelerator of cement hydration and filled in the voids during reaction.

### 3.4. Microstructural Analysis of FA Blended Cement Added α-Fe_2_O_3_ by SEM Images

SEM micrographs of FA blended cement without and with added α-Fe_2_O_3_ were used to observe the microstructure of the cement paste at 7, 28 and 90 curing days (Figure 10). With partial replacement of OPC by FA, the cement paste shows looser and more inhomogeneous microstructure with presence of less C-S-H gel (Figure 10A, point 1) and more pores (point 2). The morphology of FA blended cement with hematite nanoparticles of various curing times shows that when the curing time was longer, the C-S-H gel products formed a denser structure around the embedded FA particles. The α-Fe_2_O_3_ added mortar enhanced the compressive strength by reducing the porosity of cementitious composites, a result in accord with previous studies where iron oxides increased density [37,48,49]. The size and size distribution had the effect of fitting into available pore space in the cement composites during hydration. Moreover, α-Fe_2_O_3_ nanoparticles help the hydration reaction by increasing the nucleation of hydration products to form more C-S-H gel [40].

At 7 curing days (in the first column) all cement samples show incomplete integration of the FA particles in the overall structure of the cement reaction products. The images present many voids and incompletely covered C-S-H on FA particles. In contrast, the microstructure of the FA blended cement with added α-Fe_2_O_3_ nanoparticles consists of dense crystal hydrated products around FA according to the nucleation of α-Fe_2_O_3_ nanoparticles (Figure 10B–E). The cement paste with commercial α-Fe_2_O_3_ has C-S-H gel covered FA particles but lower than with the synthesized α-Fe_2_O_3_. Images of the synthesized α-Fe_2_O_3_ using CTAB (Figure 10D) or TX100 (Figure 10E) indicated C-S-H gel covered on FA particles at a higher level than that of synthesized α-Fe_2_O_3_ using SDS (Figure 10C).

At 28 curing days (in the second column), FA blended cement pastes were found to show two different forms including an incompletely covered C-S-H layer on FA in cement without α-Fe_2_O_3_ nanoparticles (Figure 10A) and a dense layered C-S-H on FA in cement with α-Fe_2_O_3_ nanoparticles (Figure 10B–E). These results implied that α-Fe_2_O_3_ nanoparticles acted not only as an inert material with the ability to increase the packing effect, but also led to hydration products forming around the FA. During cement hydration, the microstructure of FA blended cement with α-Fe_2_O_3_ nanoparticles becomes denser due to formation of more C-S-H gel. Especially, the microstructures of FA blended cement containing α-Fe_2_O_3_ synthesized by CTAB (D) or TX100 (E) showed a relatively densified network gel of hydration products layered on FA particles.

At 90 days (in the last column), all cement pastes were completely hydrated. The gel filling out the spaces around particles and the covering layer on FA surface can be observed. Almost all FA particles were substituted by hydrated products when FA was activated by Ca(OH)_2_. The pozzolanic reaction of FA speeded up at the later stages and the consumed content of C-S-H gel increased. The surface of FA covered by C-S-H gel were caused by pozzolanic reaction between FA and Ca(OH)_2_ and other hydration products [50]. As the process continued, the layer on the FA and cement grain thickened. The hydration products growing from the cement grains and FA are seen to be connected, although some particles still remain unreacted and acted solely as filler (Figure 10A). For FA blended cement adding α-Fe_2_O_3_ nanoparticles, the micrograph shows the surface of FA covered by C-S-H gel (Figure 10B–E). From the SEM images, more pores of the solidate cement were filled by hydration products and hematite nanoparticles. This microstructure is consistent with the enhanced compressive strength and workability. Therefore, the influence of α-Fe_2_O_3_ nanoparticles was established as a promotor of the hydration reaction, and also as a filler, leading to increased strength of cement by filling voids between FA and the hydration products.

### 3.5. Structural Model of α-Fe_2_O_3_ Nanoparticle Modified FA Cement

The hydration of cement blended with FA in this work can be illustrated as three cases: (I) without nanoparticles added, (II) with smaller and uniform particle size nanoparticles present and (III) with bigger, broadly distributed particle size nanoparticles (Figure 11). Case I (Figure 11a), the cementitious composites produce the C-S-H gel and capillary pores during hydration [51]. The added FA particles fill in the capillary pore of C-S-H gel structure to reduce the porosity in cement [52,53]. Meanwhile, as hydration progresses, the capillary pores create even more spacing that may decrease the strength of cement composites. To reduce capillary voids, α-Fe_2_O_3_ nanoparticles function as both filler and hydration accelerator that can improve the microstructure and strength of cementitious composites as Cases II and III. Case II, by adding nanosize fillers smaller than the capillary pore size, the uniform or narrow particle size distribution of nanofillers still leaves small voids from the surface-to-surface distance between the particles and consolidated cement. When the SDS synthesized α-Fe_2_O_3_ filler of smaller size and narrower particle size distribution (10–130 nm) was used, the capillary pore volume inside the cement was higher and the hydration rate was lower (Figure 11b). For Case III, large and small capillary pores are effectively filled with the bigger size and broader size distribution of nanofillers to lower the porosity in the cementitious composites. The results from hydration and cement properties of this work confirmed this hypothesis. α-Fe_2_O_3_ nanoparticles synthesized by CTAB and TX100 (range size distribution 35–195 nm) facilitated cement hydration by accelerating the formation of C-S-H gel and filling up the capillary pores, resulting in a higher density and compact microstructure (Figure 11c). With the various sizes of α-Fe_2_O_3_, the capillary pore size in cement and agglomerated particles around the cement grains are reduced according to the size of the additives that play an effective role in acceleration of hydration reaction and densification of the cement matrix [48,53,54].

## 4. Conclusions

Based on the morphologically controlled synthesis of α-Fe_2_O_3_, the spherical α-Fe_2_O_3_ of different sizes were produced using surfactant templates for nanoparticle synthesis. The smallest nanoparticles were produced when using SDS for the synthesis, whereas production using CTAB yielded the largest nanoparticles. The average particle sizes of the α-Fe_2_O_3_ synthesized via SDS and CTAB were 65.2 nm and 131.0 nm, relating to the size of their micelles. The effect of α-Fe_2_O_3_ nanoparticles on the hydration and properties of a FA cement blend was investigated as a function of concentration and size of the nanoparticles. The different nanoparticles in this study on FA blended cements illustrated how the particle size and particle size distribution of nanoparticles affect structure and properties of cementitious materials. The various particle size fractions of α-Fe_2_O_3_ nanoparticles increased the cement hydration rate and affected cementitious material pore structure development, leading to increased compressive strength. The optimum of the amount of α-Fe_2_O_3_ nanoparticles was 1.0% by weight replacement of OPC that led to the fastest hydration rate and the highest degree of hydration of cement samples. With the various sizes of synthesized α-Fe_2_O_3_ nanoparticles via CTAB and TX100, the capillary pore size in cement and agglomerated particles around the cement grains are reduced according to the size of the additives that play an effective role in acceleration of hydration reaction and densification of the cement matrix. Moreover, the α-Fe_2_O_3_ nanoparticles in binder content may help improve the overall stability and property of self-compacting concrete.

## Figures and Tables

**Figure 1 nanomaterials-11-01003-f001:**
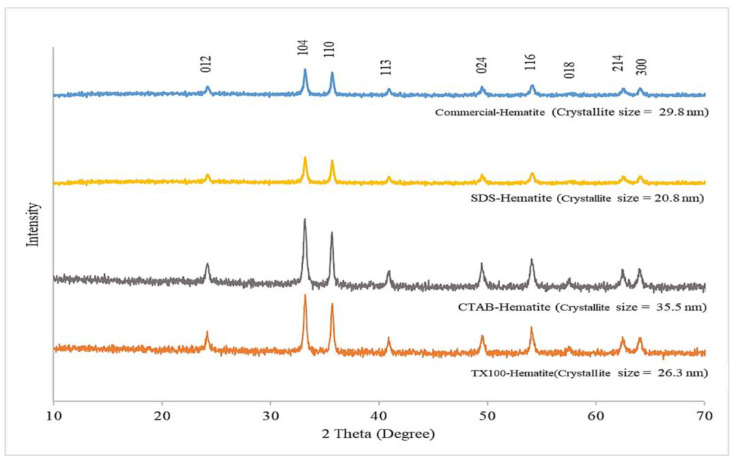
X-ray diffraction (XRD) patterns of commercial hematite (α-Fe_2_O_3_) nanoparticles and α-Fe_2_O_3_ nanoparticles synthesized using sodium dodecyl sulfate (SDS), cetyltrimethylammonium bromide (CTAB) and polyoxyethlyene tert-octylphenyl ether (TX100) as templates.

**Figure 2 nanomaterials-11-01003-f002:**
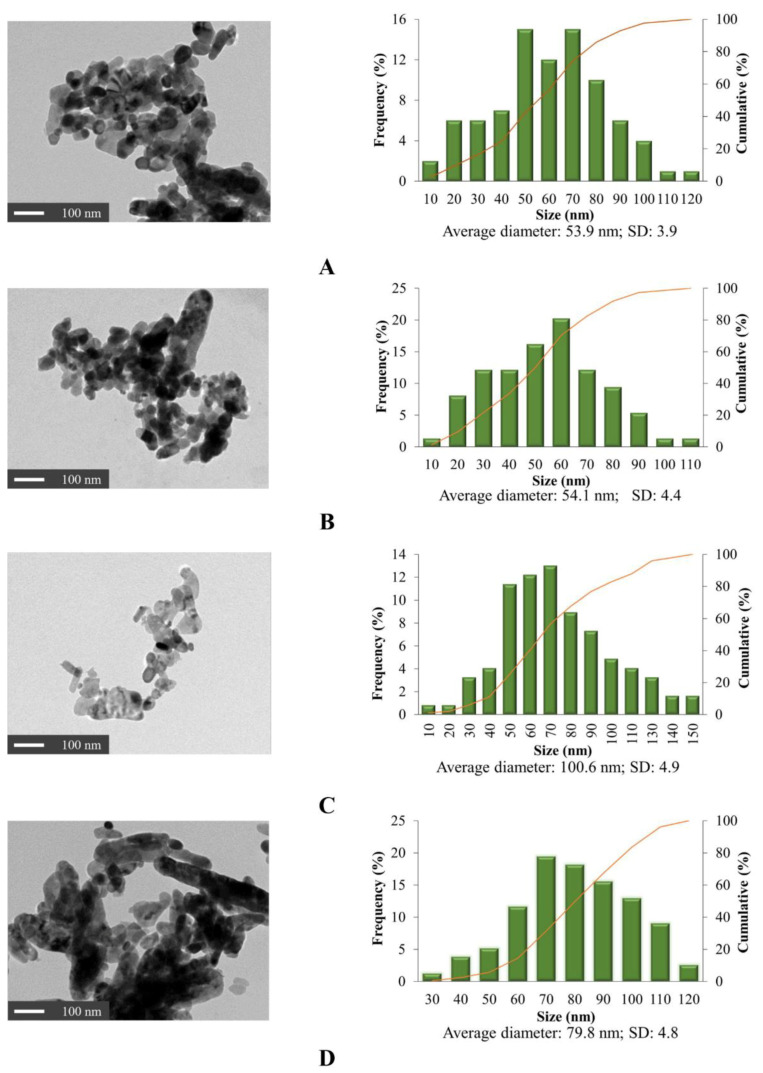
Transmission electron microscope (TEM) micrographs and grain size analysis of (**A**) commercial α-Fe_2_O_3_, (**B**) synthesized α-Fe_2_O_3_ at 2× critical micelle concentration (CMC) SDS, (**C**) synthesized α-Fe_2_O_3_ at 2× CMC CTAB and (**D**) synthesized α-Fe_2_O_3_ at 2× CMC TX100.

**Figure 3 nanomaterials-11-01003-f003:**
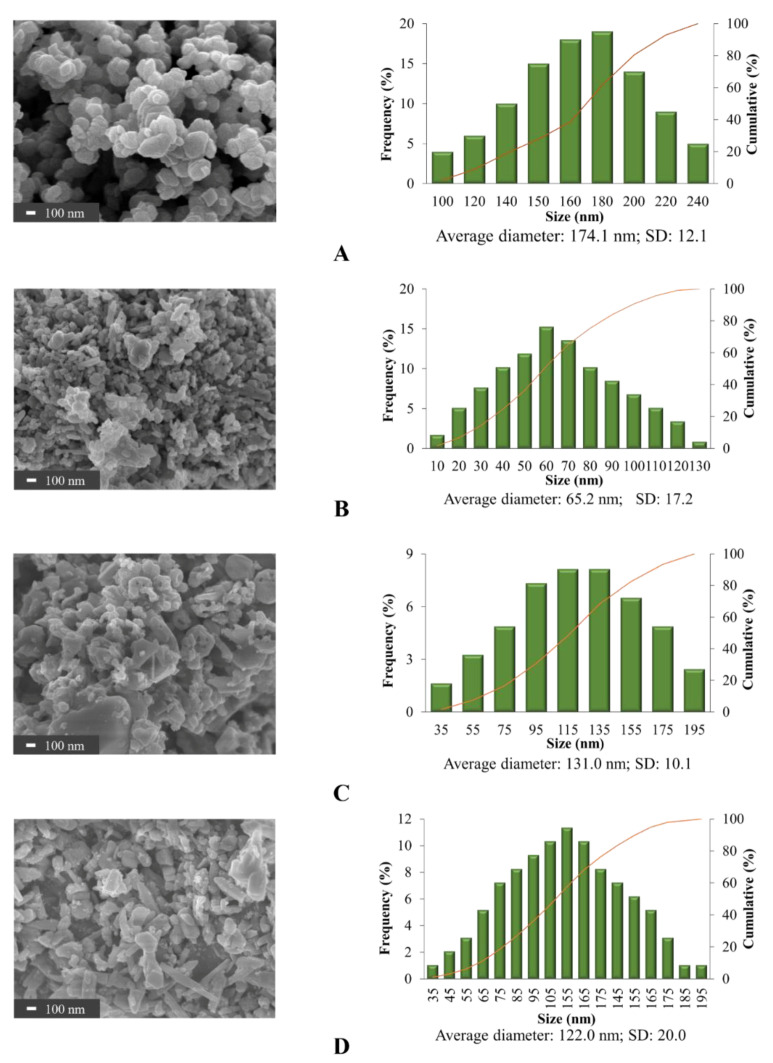
Morphology by field-emission-scanning-electron-microscopy (FE-SEM) imaging by secondary electron detector and size analysis of (**A**) commercial α-Fe_2_O_3_, (**B**) synthesized α-Fe_2_O_3_ at 2× CMC SDS, (**C**) synthesized α-Fe_2_O_3_ at 2× CMC CTAB and (**D**) synthesized α-Fe_2_O_3_ at 2× CMC TX100.

**Figure 4 nanomaterials-11-01003-f004:**
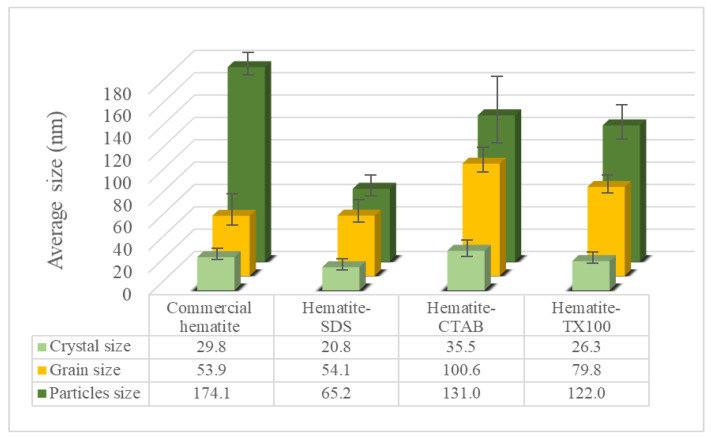
Crystal size obtained by XRD, grain size from ImageJ analysis of TEM images and particle size from ImageJ analysis of FE-SEM images of α-Fe_2_O_3_ nanoparticles.

**Figure 5 nanomaterials-11-01003-f005:**
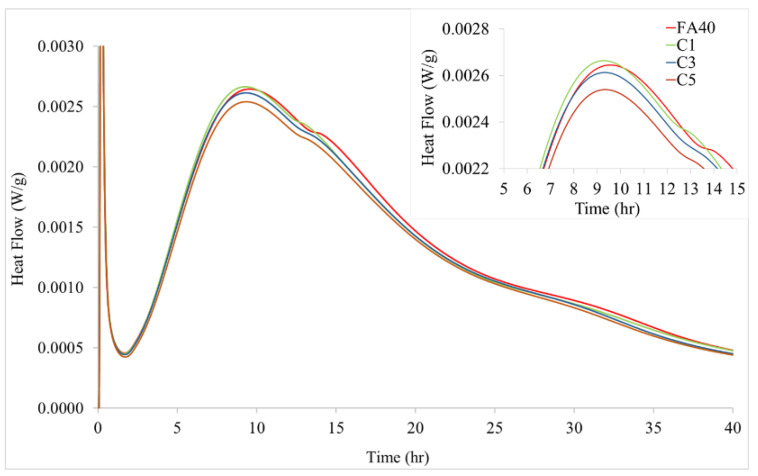
Effect of amount of commercial α-Fe_2_O_3_ at 0.0%, 1.0%, 3.0% and 5.0% on the time dependent heat of hydration for fly ash (FA) blended cement. Inset: Enlarged image for heat flow during stages 3 and 4 to show peak maxima for samples C1-C3 and control FA40.

**Figure 6 nanomaterials-11-01003-f006:**
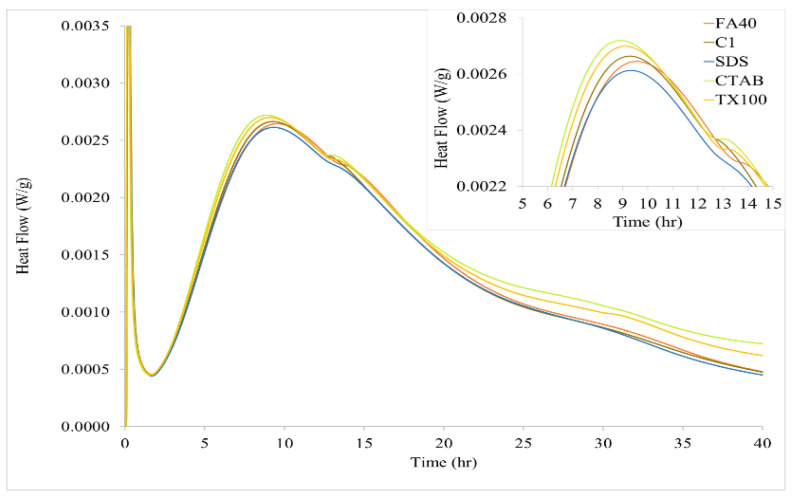
Effect of various synthesized α-Fe_2_O_3_ nanoparticles (1.0%) on the heat of hydration of FA blended cement. Inset: Enlarged image for heat flow during stages 3 and 4 showing peak maxima for control FA40 and nanoparticle samples.

**Figure 7 nanomaterials-11-01003-f007:**
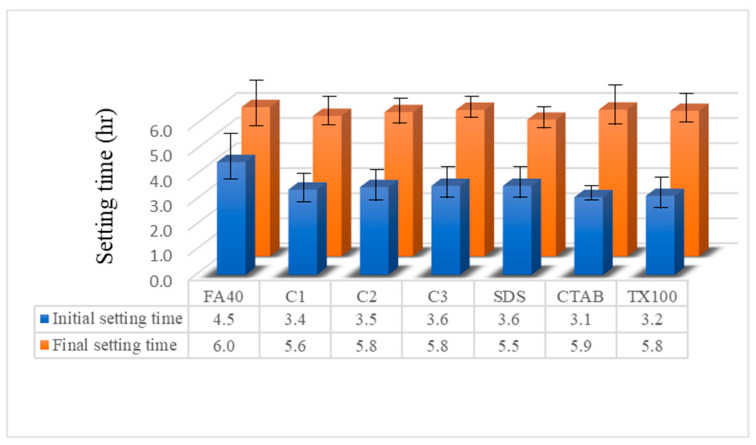
Effect of concentration and synthesized α-Fe_2_O_3_ nanoparticles on setting time of FA blended cement mortar.

**Figure 8 nanomaterials-11-01003-f008:**
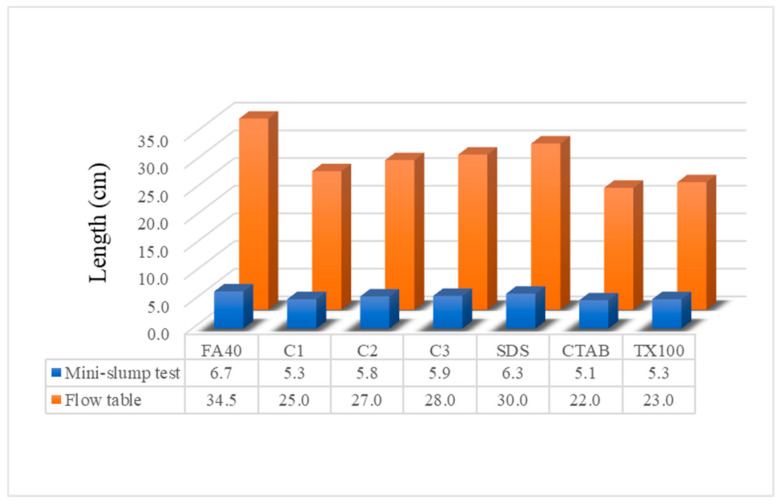
Effect of α-Fe_2_O_3_ nanoparticles on mini-slump test and flow table of FA blended cement mortar.

**Figure 9 nanomaterials-11-01003-f009:**
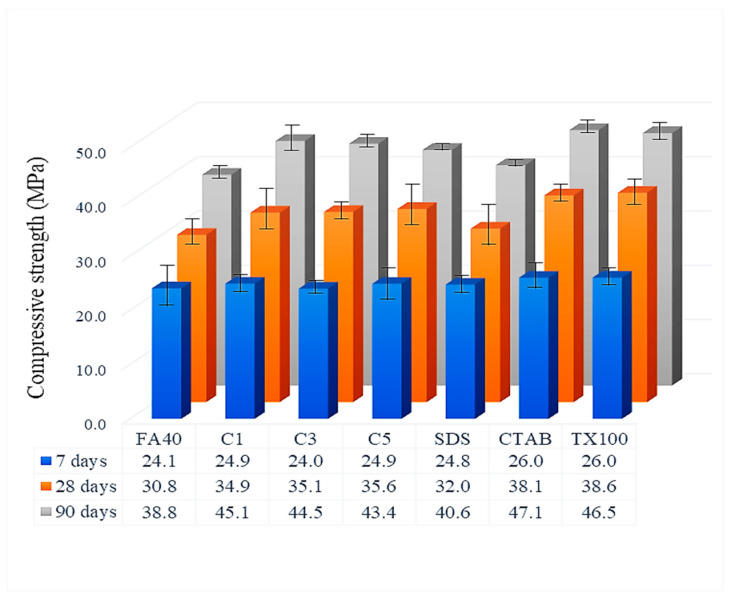
Compressive strength of FA blended cement mortar at 7, 28 and 90 days via varying α-Fe_2_O_3._

**Figure 10 nanomaterials-11-01003-f010:**
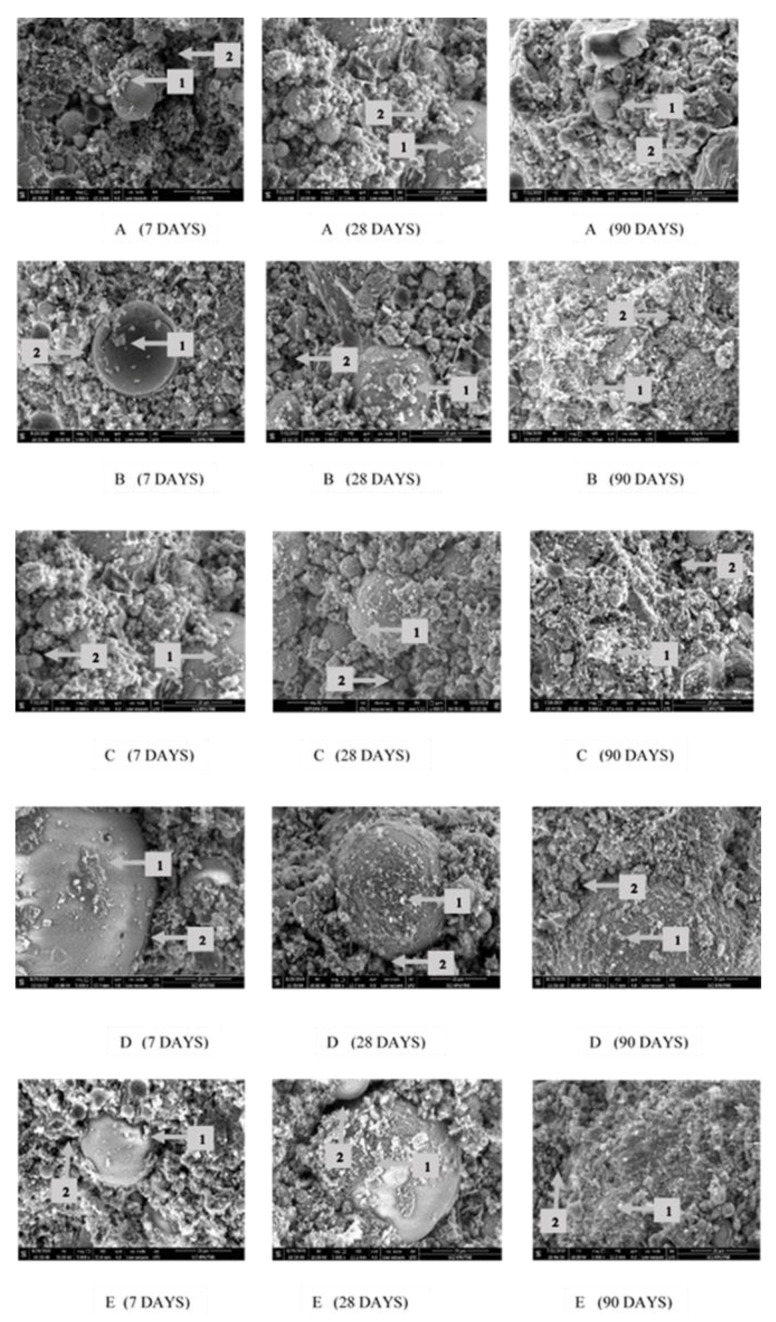
SEM micrographs of hydrated FA blended cement at 7, 28 and 90 days showing layered Ca(OH)_2_ around FA (point 1) and the air voids (point 2): varying α-Fe_2_O_3_ of (**A**) FA40, (**B**) C1, (**C**) SDS, (**D**) CTAB and (**E**) TX100.

**Figure 11 nanomaterials-11-01003-f011:**
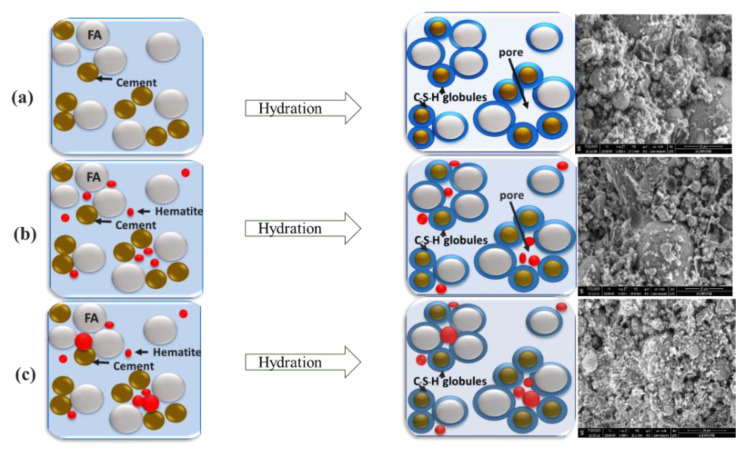
Schematic of cement hydration of (**a**) no α-Fe_2_O_3_, (**b**) α-Fe_2_O_3_ in the narrow range particle size distribution and (**c**) α-Fe_2_O_3_ in the broad particle size distribution.

**Table 1 nanomaterials-11-01003-t001:** The fraction of OPC, fly ash and α-Fe_2_O_3_ nanoparticles as a binder and their codes.

Sample Name	Proportions		
	OPC	Fly Ash	Amount and Type of α-Fe_2_O_3_
FA40	60%	40%	No α -Fe_2_O_3_ used as reference
C1	59%	40%	1.0% commercial α-Fe_2_O_3_
C3	57%	40%	3.0% commercial α-Fe_2_O_3_
C5	55%	40%	5.0% commercial α-Fe_2_O_3_
SDS	59%	40%	1.0% α-Fe_2_O_3_ synthesized with SDS
CTAB	59%	40%	1.0% α-Fe_2_O_3_ synthesized with CTAB
TX100	59%	40%	1.0% α-Fe_2_O_3_ synthesized with Triton X 100

## Data Availability

The data that support the findings of this study are available from the corresponding author upon reasonable request.

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
