# Peer review of "Effect of Morphologically Controlled Hematite Nanoparticles on the Properties of Fly Ash Blended Cement"

_nanomaterials, 2021, doi:10.3390/nano11041003_

Round 1
Reviewer 1 Report
I suggest that the paper is clearly presented and the methods and results are well described.

Author Response
Thank you for your consideration on our manuscript. As your suggestion in some points, we have improved and answered as attached file.

Reviewer 2 Report
Article titled: Effect of morphologically controlled hematite nanoparticles on the properties of fly ash blended cement, presents very interesting investigation. In my opinio it can be publshed after some fullfilments and improvements.
In the article authors synthesized nanoparticles of hematite, hovewer there is no explanation target of this activities. I mean economic aspects? There is lack of information about even appoximately estimation of costs synthesis especially that cost of proces with usage of reagents such as CTAB are high.
Second isse why authors conisde maxiumum period of time for ageing only 90 days. Most of investigation presents in literature carreid out some test also after one year. In such case authors should mention in conclusion chapter about further plans.
And Conclusion constitued rather summary rather than main conslusion, therefore it sould be improved.
Author Response

(The authors gave the same response as above.)

Reviewer 3 Report
Some of the Figures need improvement, see comments in the pdf

Author Response

(The authors gave the same response as above.)
